Rapid multi-locus sequence typing direct from uncorrected long reads using Krocus

http://orcid.org/0000-0001-6919-6062 Page Andrew J. 1 2 andrew.page@quadram.ac.uk
http://orcid.org/0000-0003-0739-6848 Keane Jacqueline A. 2
1 Quadram Institute Bioscience , Norwich Research Park, Norwich , UK
2 Pathogen Informatics, Wellcome Sanger Institute , Hinxton, Cambridgeshire , UK
Rahmann Sven
Electronic publication date: 2018 Jul 31
Publication date: 2018
Volume: 6
Electronic Location ID: e5233
Received 2018 Mar 20; Accepted 2018 Jun 25
Copyright: © 2018 Page and Keane
Copyright year: 2018
Copyright holder: Page and Keane
License: This is an open access article distributed under the terms of the Creative Commons Attribution License, which permits unrestricted use, distribution, reproduction and adaptation in any medium and for any purpose provided that it is properly attributed. For attribution, the original author(s), title, publication source (PeerJ) and either DOI or URL of the article must be cited.
License URL: https://creativecommons.org/licenses/by/4.0/

Keywords: Microbial, Bioinformatics, Nanopore, Pacbio, Multi-locus sequence typing

Funding: Quadram Institute Bioscience BBSRC funded Core Capability Grant project number BB/CCG1860/1 Wellcome Trust WT 098051 This work was supported by the Quadram Institute Bioscience BBSRC funded Core Capability Grant (project number BB/CCG1860/1) and by the Wellcome Trust (grant WT 098051). The funders had no role in study design, data collection and analysis, decision to publish, or preparation of the manuscript.

==============================
Genome sequencing is rapidly being adopted in reference labs and hospitals for bacterial outbreak investigation and diagnostics where time is critical. Seven gene multi-locus sequence typing is a standard tool for broadly classifying samples into sequence types (STs), allowing, in many cases, to rule a sample out of an outbreak, or allowing for general characteristics about a bacterial strain to be inferred. Long-read sequencing technologies, such as from Oxford Nanopore, can produce read data within minutes of an experiment starting, unlike short-read sequencing technologies which require many hours/days. However, the error rates of raw uncorrected long read data are very high. We present Krocus which can predict a ST directly from uncorrected long reads, and which was designed to consume read data as it is produced, providing results in minutes. It is the only tool which can do this from uncorrected long reads. We tested Krocus on over 700 isolates sequenced using long-read sequencing technologies from Pacific Biosciences and Oxford Nanopore. It provides STs for isolates on average within 90 s, with a sensitivity of 94% and specificity of 97% on real sample data, directly from uncorrected raw sequence reads. The software is written in Python and is available under the open source license GNU GPL version 3.

Introduction

With rapidly falling costs, long-read sequencing technologies, such as from Pacific Biosciences (PacBio) and Oxford Nanopore Technologies (ONT), are beginning to be used for outbreak investigations (Faria et al., 2017; Quick et al., 2015) and for rapid clinical diagnostics (Votintseva et al., 2017). Long-read sequencers from Oxford Nanopore can produce sequence reads in a matter of minutes and sequencers from PacBio can produce sequences in a number of hours compared to short-read sequencing technologies which takes hours/days. Seven gene multi-locus sequence typing (MLST) is a widely used classification system for categorising bacteria. It can be used to quickly rule an isolate out of an outbreak and knowing a sequence type (ST) can often allow for general characteristics of a bacteria to be inferred. By reducing the time from swab to an actionable answer, genomics can begin to directly influence clinical decisions, with the potential to make a real positive impact for patients (Gardy & Loman, 2018).

With the increased speed afforded by long-read sequencing technologies comes increased base errors rates. The high error rates inherent in long-read sequencing reads require specialised tools to correct the reads (Koren et al., 2017), however, these methods have substantial computational resource requirements often taking longer to run than the original time to generate the sequencing data.

A full overview of MLST software for short-read sequencing technologies is available in Page et al. (2017). Of the software reviewed in Page et al. (2017), only the methods which take a de novo assembly as input can be used with long-read sequencing technologies; however, de novo assembly has a substantial post processing computational overhead, which can exceed the time taken to perform the sequencing. StringMLST (Gupta, Jordan & Rishishwar, 2017), was designed to rapidly predict MLST from raw read sets by performing a k-mer analysis. MentaLiST (Feijao et al., 2018) takes a similar k-mer analysis approach and is designed for large typing schemes, such as cgMLST and wgMLST. They were designed to work only with high base quality short-read sequencing data. To our knowledge, no method currently exists for calling MLST from uncorrected long-read sequencing data.

We present Krocus which can rapidly estimate STs directly from uncorrected long reads from isolates. Results are presented using the largest public dataset of bacterial long-read data containing over 700 samples generated using the PacBio sequencing technology, and for real and simulated datasets of ONT data. On average it produces sequence correct STs in 90 s, with a sensitivity of 94% and specificity of 97% for uncorrected PacBio sequence data. On a dataset consisting of 524 simulated ONT samples based on E. coli reference genomes the sensitivity was 82–94% depending on the flow cell modelled. On a small real ONT dataset of 12 Klebsiella pneumoniae a sensitivity of 100% was achieved. Krocus is the only tool which can call MLST directly from uncorrected long reads with high accuracy. It is written completely in Python 3 and is available under the open source licence GNU GPL 3 from https://github.com/andrewjpage/krocus.

Materials and Methods

The basic method of Krocus is to take short k-mers and calculate the coverage over the MLST alleles and a flowchart showing the method is presented in Fig. 1. As the base errors are mostly uniformly distributed, a well chosen k-mer value results in short stretches of error free bases. Some k-mers will be erroneously flagged due to errors however as more reads are added (above 5×), these errors are filtered out as they have a low occurrence overall.

Figure 1 Flowchart of the Krocus method.

The square boxes denote processes that the data undergoes, the diamond shapes denote a decision point, the box with a wavy lower line denotes a data file, and the tabs denote an output.

Krocus takes as input the path to an MLST scheme, a FASTQ file containing uncorrected long reads from an isolate and a k-mer size. The MLST alleles are contained in seven FASTA files, downloaded from PubMLST (Jolley & Maiden, 2010) or taken from the set distributed with the software. Each sequence in the allele files contains a unique identifier and the combination of these allele identifiers gives rise to the ST, contained within a profile tab delimited file. An alignment-free k-mer sequence analysis approach is used to determine the presence and absence of particular alleles, with optimisations for high error rate long-read sequencing data. For a given k-mer size, the k-mers are extracted from each sequence in each allele file.

In long-read sequencing reads, whilst there are high base error rates, the errors are mostly uniformly distributed. The ideal k-mer size is the mean number of bases on a read which is free from errors, for example if the base error rate n is 91% an error occurs on average every ∼11 bases, thus the k-mer size k calculated as k = ⌊100/(100−n)⌋. A k-mer size which is too high would invariably always include an erroneous base, reducing the probability of a match with the allele files. A k-mer size which is too low would reduce the possible k-mer space and lead to an increase in matches by random chance. Each long read is inspected and k-mers are generated with a step size of k giving an average depth of k-mer coverage of 1. If a single k-mer from this set is present in the allele k-mers, the read is kept for further analysis, if no k-mers are present, the read is discarded as it is unlikely to contain the MLST genes.

All possible k-mers are generated for the read which passed the initial filtering with a step size of 1, giving an average depth of k-mer coverage equal to k, with k-mers occurring more than five times excluded from further comparison as they do not impart useful information. For each allele file, the intersection of the allele k-mers and the read k-mers is taken. The read is split up into bins of length k, and the intersecting k-mers are assigned to their corresponding bin in the read to produce an approximate k-mer coverage of the read. A sliding window (default four times k-mer size) is used to span short gaps, which are likely the result of small errors in the underlying sequencing data. The largest contiguous block of k-mer coverage in the read is identified, based on the sliding window results, and if it meets the minimum block size (default 150 bases, derived from ⅓ of the average length 467 of all sequences in PubMLST, retrieved 02-02-18), it is said to contain one of the typing alleles. The block is extended by 100 bases on either side to ensure the full allele is captured. The k-mers matching this block in the read are extracted and k-mer counts corresponding to the allele matching k-mers are incremented. The read is reverse complemented and the same search is undertaken once more.

At defined intervals (default 200 reads) the genes of each allele are analysed to calculate the number of k-mers covered by the raw read, allowing for the input files to be streamed in real-time as data is generated. Only ONT sequencers support real-time streaming of reads. PacBio sequencers require post analysis once the sequencing experiment is complete. If an allele has a gene with 100% k-mer coverage, it is said to be present, if it is less than 100%, the allele with the most number of k-mers is identified, but with a low confidence flag. Where two or more alleles of the same gene have 100% coverage, the sequence with the highest k-mer coverage is used. Novel combinations of alleles and new, unseen, alleles cannot be reliably detected using this method, and so are excluded from the analysis.

PacBio samples

The NCTC 3000 project (http://www.sanger.ac.uk/resources/downloads/bacteria/nctc) aims to produce 3,000 bacteria reference sequences using the PacBio long-read sequencing technology. Each of the reference strains was selected for sequencing to maximise diversity and to capture historically medically important strains. This is currently the largest public long-read sequencing project for bacteria and is still on-going with 1,048 assemblies publicly available (accessed 2/1/18). The assemblies were downloaded from the project website and the sequencing reads directly from the European Nucleotide Archive. The sequencing reads were all generated on the PacBio RSII between 2014 and 2017. The assemblies used for comparison with Krocus were generated using an open source pipeline (https://github.com/sanger-pathogens/vr-codebase) which first performed a de novo assembly using HGAP (SMRT analysis version 2.3.0) (Chin et al., 2013), followed by circularisation with circlator (version 1.5.3) (Hunt et al., 2015), and finally automated polishing with the resequencing protocol (SMRT analysis version 2.3.0) from PacBio.

Each assembly (1,048) was ST using the TS-mlst software (https://github.com/tseemann/mlst). Any samples identified as ambiguous or untypable by TS-mlst were excluded from further analysis. Also, samples where there were no corresponding MLST scheme for the species (339) were also excluded, as a meaningful comparison cannot be made. The TS-mlst software was shown in Page et al. (2017) to never make any false positive ST calls. The remaining 709 samples are detailed in Table S1, including accession numbers, and summarised in Table 1, covering 43 species and 638 STs with representatives from both gram positive and gram negative. The FASTQ files of the uncorrected reads were generated from the raw data using the PacBio SMRTlink pipeline (version 5.0.1.9585), and the time for this conversion is not considered in the results presented in this paper as it is a standard preprocessing step required for many downstream analyses.

Table 1 NCTC 3000 PacBio sequenced samples with results after analysis with Krocus.

Species	No. of samples	No. of unique STs	No. in agreement	Mean wall time (s)	Mean reads	
E. coli	226	204	204	102	17,524	
E. faecalis	11	10	10	42	19,900	
K. pneumoniae	113	101	109	62	22,297	
P. aeruginosa	22	21	18	127	41,444	
S. aureus	114	92	111	122	16,255	
S. dysgalactiae	16	16	16	32	10,412	
S. enterica	48	46	47	107	16,348	
S. pyogenes	47	47	47	37	7,714	
Other	112	101	106	57	15,024	
Total	709	638	668	86	17,439	
Note:

An ST is said to be in agreement if it matches the ST called by TS-mlst from a de novo assembly.

PacBio control samples

A set of 74 samples representing 48 species were selected as controls from the NCTC 3000 project. Each were sequenced using the PacBio long-read sequencing technology as described previously and are listed in Table S2. The controls were selected from within the same genus as the cases as listed in Table 1, but from different species. The species classifications came from experimental techniques. The de novo assemblies of each sample were analysed with the TS-mlst software, and any which resulted in valid STs were removed to reduce the impact of confounders from misclassified isolates.

ONT samples

To analyse the performance of Krocus on ONT data, we selected a set of 12 K. pneumoniae samples used previously for performance comparisons (Wick, Judd & Holt, 2018; Wick et al., 2017b). The dataset is available from Wick (2017a) and detailed in Table S3. For comparison Canu (Koren et al., 2017) (version 1.5) assemblies and Unicycler (Wick et al., 2017a) (version 0.4.0) assemblies, utilising Miniasm (Li, 2016) and Racon (Vaser et al., 2017), post Nanopolish (https://github.com/jts/nanopolish) (version 0.7.0), created using only the long-read data (Wick, 2017b) were used.

There are currently no large publicly available ONT datasets. To overcome this, simulated ONT reads were generated using NanoSim-H (https://github.com/karel-brinda/NanoSim-H) (version 1.1.0.2), a derivative of NanoSim (Yang et al., 2017). NanoSim-H includes error models for simulating E. coli uncorrected nanopore reads with multiple flow cell types. Every complete E. coli reference genome in RefSeq was downloaded (n = 549). TS-mlst was run on these reference genomes to generate the target ST. TS-mlst was unable to call an E. coli ST in 25 cases and these were excluded from further analysis. Of the samples, 11 contained one or more incomplete gene, three contained duplicated genes, two were missing a gene, and one was a novel combination of alleles. Extended details of the samples and the results are listed in Table S4. Using NanoSim-H 20,000 reads were simulated with the R9 flow cell error model (R9_1D and R9_2D) and the default parameters for each reference genome (n = 524). The reads were converted from FASTA format to FASTQ format using PyFASTAQ (https://github.com/sanger-pathogens/Fastaq) (version 3.17.0) and provided to Krocus (default parameters).

Compute resources

All experiments on real datasets were performed using the Wellcome Sanger Institute compute infrastructure running Ubuntu 12.04 LTS, with each host containing 32 cores (AMD Opteron Processor 6272) and 256 GB of RAM. All experiments involving simulated datasets were performed using the MRC CLIMB (Connor et al., 2016) cloud infrastructure running the Genomics Virtual Lab (Afgan et al., 2015), with each host containing eight cores and 64 GB of RAM. Only a single core was used in each performance experiment with the mean memory requirement of 0.354 GB (std dev 0.16).

Results

PacBio results

Each of the assemblies from the NCTC dataset were run through TS-mlst to generate a ST. Krocus was run for each sample using the uncorrected FASTQ files and default settings and halted when the ST matched the expected result from TS-mlst. The time to achieve the correct predicted ST was noted, as were the number of reads, with a mean of 86 s, after processing a mean of 17,439 reads. The number of reads required before Krocus correctly predicts the ST is presented in Fig. 2A, with only species with 10 or more isolates included. The running time for each species is presented in Fig. 2B. The running time of three Staphylococcus aureus samples was elevated due to the need to process a higher than average number of reads; however, within 60 s six out of seven alleles had been correctly identified, with the last allele taking up to a further 11 min to identify correctly. In 94% of cases (sensitivity) the results from Krocus and TS-mlst were in agreement, with the calculations listed in Table S5.

Figure 2 The reads and time to correctly predict an ST for each PacBio NCTC species analysed.

(A) Number of reads analysed before the Krocus correctly predicted an ST for each PacBio NCTC species analysed. (B) Time in seconds before Krocus correctly predicted an ST for each PacBio NCTC species analysed.

In 43 cases (6%) STs did not match the expected ST or were untypable, with 40 of these calling six out of seven typing genes correctly. In the remaining one case, five out of seven genes were called correctly. These failures are due to known systematic errors with long homoployers with the PacBio sequencing technology (Quail et al., 2012) which cannot be overcome with short k-mers.

The control samples were analysed in a similar fashion to give a specificity of 97% (72 out of 74). In the two false negative cases, both contained all seven typing genes, with one containing two copies of gene phoE which Krocus was unable to distinguish, and one containing a variant in fumC which was not in the typing database.

ONT results

For all 12 K. pneumoniae samples (100%) of uncorrected ONT reads Krocus provided the expected ST. The mean time to the expected ST was 134 s after a mean of 3,250 reads. As a comparison, de novo assembled genomes using the ONT reads alone from Wick et al. (2017b) did not identify any of the STs when analysed by TS-mlst. This was due to the inherent high base error rate which resulted in a poor quality assembly. Only hybrid assemblies additionally utilising Illumina short-read data could be ST. This gives Krocus an advantage over de novo assembly of ONT only reads.

For the simulated E. coli reference genome reads, two flow cell models were used, R9 1D and R9 2D. The lower quality R9 1D simulations correctly identified the STs in 432 (82.4%) cases with a mean running time of 71 s and a mean of 6,618 reads. Of the 92 STs which were not called correctly, 84 were as a result of errors in calling fumC or gyrB. The higher quality R9 2D simulations correctly identified STs in 492 (93.8%) cases with a mean running time of 25 s and after a mean of 5,894 reads. Of the 36 STs which were not called correctly, 34 were as a result of errors in calling fumC or gyrB. The difference in running time is due to lower error rates in the R9 2D uncorrected reads, reducing the number of observed k-mers. The quality of the base called data used for generating the simulation models directly impacts the ability to call STs accurately.

Discussion

As shown previously it is possible to call STs directly from uncorrected long-read data. Whilst the error rates in uncorrected reads are high, the error profile of long reads is such that short regions of high quality data exist between the errors in the reads. Read correction typically overlaps reads and calls a consensus to fix errors in the underlying read. The ability to utilise uncorrected reads and still generate accurate results means that time consuming read correction steps are not needed to generate sequence typing information. Krocus achieves this by using short k-mers.

The PacBio sequencer produces uncorrected reads in both BAM format and HDF5 format (legacy). These are not available to call in real-time due to the requirement for post-sequencing base calling, so Krocus would only ever be run on the final data at the end of a sequencing run. The ONT sequencers can produce uncorrected reads in FASTQ format in real-time. These can be streamed directly into Krocus taking maximum advantage of the method. Additionally, as the ONT sequencers can be halted mid-run, flushed and loaded with a new sample, a single flow cell can be reused many times over, potentially as soon as Krocus has generated a ST. This has the effect of reducing the costs of the sequencing run.

Due to a current lack of large ONT datasets in the public archives, simulated data was used. Simulated ONT reads gives an insight into the performance of Krocus with different error models, with more accurate read sets resulting in more accurate ST calling. The ONT sequencing technology and the base calling software are currently undergoing rapid change with constant improvements in the quality of the data, so these simulations should seen as a baseline performance for Krocus.

Conclusion

Krocus is the only tool which can call STs directly from uncorrected long reads with high accuracy. The sensitivity of 94% and specificity of 97% achieved by Krocus on a large, diverse, PacBio dataset is similar to gold standard experimental standard methods (Liu et al., 2012). By calling STs directly from uncorrected long reads, the need for post processing steps and de novo assembly is eliminated, reducing the turnaround time for MLST from days to minutes. For a small ONT dataset of real samples, Krocus correctly called the ST in all cases, compared to de novo assemblies of the same data, where no STs could be called. In a large simulated ONT dataset of E. coli, the sensitivity was 82–94%, depending on the flow cell modelled.

Supplemental Information

Supplemental Information 1 Supplementary tables listing accession numbers and detailed experimental results.

Click here for additional data file.

We wish to thank Nick Grayson from the Wellcome Sanger Institute for assistance with the NCTC 3000 dataset. Thanks to João Carriço and Nabil-Fareed Alikhan for providing helpful feedback and suggestions for this paper. Thank you to Karel Břinda and Nick Loman for reviewing this paper and providing helpful feedback.

Additional Information and Declarations

Competing Interests

Author Contributions

Data Availability

The authors declare that they have no competing interests.

Andrew J. Page conceived and designed the experiments, performed the experiments, analysed the data, contributed reagents/materials/analysis tools, prepared figures and/or tables, authored or reviewed drafts of the paper, approved the final draft.

Jacqueline A. Keane authored or reviewed drafts of the paper, approved the final draft.

The following information was supplied regarding data availability:

GitHub: https://github.com/andrewjpage/krocus.

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
