# Peer review of "Rapid multi-locus sequence typing direct from uncorrected long reads using Krocus"

_PeerJ, doi:10.7717/peerj.5233_

## Round 0.1 · original submission · Major Revisions

As you can see, your manuscript has been seen by three experts who have written reviews at different levels of detail and hence ask different questions. May I suggest that you try to reproduce and resolve the difficulties that reviewer 1 (Karel Břinda) encountered, answer each question from each reviewer and add the requested clarifications.

·

Basic reporting

No comment

Experimental design

Page & Keane present Krocus, a k-mer-based method for a rapid detection of Multilocus Sequence Types from long reads produced by Pacbio and Oxford Nanopore sequencing technologies. MLST is a standard technique in molecular biology used for classifying strains of a species into smaller groups, based on the observed alleles of a tuple of housekeeping genes. Being originally designed as a PCR-based technique, MLST remained widely popular even in the era of NGS. While several MLST callers exist for calling from assemblies or short read sequencing technologies, no program exists for long reads. The paper is thus highly relevant and Krocus is likely to find its use in practical applications.

Krocus uses a k-mer-based approach. It iterates over a stream of reads. For each read, it starts by filtering, looking for alleles that might have a match within a given read. Then the candidate alleles are compared to the read and shared k-mers identified. This is followed by a correction for sequencing errors. The reconstructed k-mers hits and read coverage are then used to update counters corresponding to the allele and reporting the identified alleles and the corresponding sequence type.

All steps of the program are well explained. According to my testing, Krocus is easy to use. Its source codes are available from Github, tested on Travis CI, released under the GNU GPL 3 license, and they are well documented and easy to read and understand. The Krocus Python package can be installed using PIP from Github. Detailed data about individual experiments are provided in an Excel spreadsheet, which contains accession numbers of the tested samples, true sequence types, predicted sequence types, and other information.

Validity of the findings

According to the authors, Krocus is highly accurate and sensitive (94% sensitivity, 97% specificity) and provides results within 90 seconds from the start of sequencing (average). To validate these characteristics, I performed several experiments with own data.

Unfortunately, despite extensive experimenting with the Krocus CLI parameters, I did not succeed in reproducing the presented characteristics. In particular, I tested Krocus with six isolates of Streptococcus Pneumoniae sequenced using Oxford Nanopore. In only one case the correct sequence type (ST) was reliable detected. Here is a summary of the individual cases:
1) ST 62: Unsuccessful; oscillations between the correct and a novel ST
2) ST 320: Unsuccessful; oscillations between an incorrect (9630) and a novel ST
3) ST 37: Unsuccessful; unstable several stretches of novel STs detected (spi gene: spi(425), spi(13), spi(381), …)
4) ST 320: Unsuccessful; stable, but an incorrect ST detected (9630)
5) ST 3587: Unsuccessful; no ST detected
6) ST 37: Successful; stable and a correct ST detected

The speed of detection (90s) couldn’t be reproduced too. In most of the experiments, the results were not stable and even after several hours of (simulated) sequencing – Krocus was still switching between different STs.

Additional comments

The source of the discrepancy between my results and the results from the paper is not fully clear to me. It might be consequence of a slightly limited robustness of the method and of the dependence on multiple hardcoded and user-specified parameters. These parameters seem to be calibrated for the high-quality NCTC data, and probably only to those species that are considered in the comparisons. This might not be so problematic if the failures were clearly indicated by Krocus. Unfortunately, even though the program provides a mechanism to indicate low confidence allele calls (using the star character), these warnings were not reliable (many false negatives).

I see two possible solutions. Either the paper can be modified so that it becomes clear under which assumptions Krocus is guaranteed to work (e.g., enumerated species + only PacBio + error rate lower than a certain threshold), or the method should be improved so that it can work for both ONP and PacBio in a general setting. It would also be nice if Krocus could clearly indicate failures (see Specific comments). I also suggest using simulated reads to evaluate the effect of sequencing errors on MLST calling and reliability of the method for all supported species.

Specific comments

• The assumptions for the method to work should be well documented. Does Krocus work only with isolates, or it supports also metagenomes? Is any prior cleaning of the reads needed or recommended for increasing the quality of the reported MLST calls? Is Krocus supposed to work with all the species that are listed in the output of “krocus_database_downloader -l”?
• It is not clear to me how to decide whether the current result is final or not (when looking at the stream of intermediate outputs). Krocus is often oscillating between multiple sequence types, and it also tends to report a novel sequence type too often. There is no number in the output, which would help to distinguish between reliable and unreliable detections (an equivalent of mapping quality in read mapping). Could Krocus report some kind of confidence score?
• Would it be possible to upload Krocus to Bioconda and PyPI?
• It is not clear why certain sequencing experiments from NCTC 3000 are excluded from the comparison (l.121). These examples might help to explain why Krocus failed in my tests (e.g., whether this behavior is species-specific).
• It would be nice to provide a simple figure illustrating the algorithm (the bins, etc.; l.70-103).
• I would highly appreciate a Make/Snakemake pipeline for downloading the NCTC samples and re-runing the experiments.

Reviewer 2 ·

Basic reporting

no comment

Experimental design

no comment

Validity of the findings

no comment

Additional comments

This manuscript introduces Krokus, a software tool to infer MLST sequence types from un-assembled sequence data from sequence error-rich, long read sequence platforms such as PacBio or Oxford Nanopore. The authors explain the algorithm in detail, test its accuracy as compared to another WGS MLST tool. The paper is generally well written, the experiments are well designed, and the findings valid. While well written, the description of the algorithm would benefit from an additional figure laying out the individual steps of the algorithm.

Reviewer 3 ·

Basic reporting

The writing is clear and unambiguous, correctly referenced, with good quality figures and tables and availability of code and data.

Experimental design

This paper describes a novel bioinformatics pipeline which is open source and well-documented.

Validity of the findings

No concerns on validity.

Additional comments

The authors describe a new bioinformatics tool and algorithm, Krocus, for identifying MLST profiles from single molecule reads generated from nanopore sequencing and PacBio.

MLST profiling remains an important tool in public health microbiology to give subtype designations to bacterial species.

The software is available as an open source package with good documentation.

The tool is fast to use and has good performance, except in a small percentage of cases, which is ascribed to sequencing error.

I was able to install the software easily on Linux and Mac, download a database, and test a nanopore FASTA file which rapidly gave me the correct ST. Congratulations!

The tool has a few limitations - it can only be used on isolate data (not metagenomes) and cannot be used to reconstruct novel alleles.

My comments are minor:

1. I would say you cannot really use MLST to rule an isolate into an outbreak, only to rule out. Finer grained SNP or cgMLST analysis is commonly used to infer whether isolates are from a point source. Does this method also work for cgMLST?

2. I actually have doubts that this method is definitively better/faster than an alignment-based approach. Have the authors simply attempted to map reads to a reference MLST database and count the top hits using a fast mapper like minimap2? I suspect this would give similar results to this approach (in reality, the approach used is not that dissimilar to accelerated alignment methods). I don’t think a comparison is necessary in this manuscript, so please remove references to this approach being a better method than alignment-based ones (or alteranatively do the comparison).

3. 145-145 I think what are referred to as Unicycler+nanopolish assemblies are a pipeline that use miniasm + Racon, followed by nanopolish. This should be clarified (and references cited). It may be useful to check results of TS-mlst on Canu+nanopolish assemblies as well.

4. PacBio, although a real-time sequencing instrument, does not currently provide read data in real-time for analysis. Instead a video is recorded for a fixed time, then basecalled and subsequently analysed. By contrast, nanopore reads are available for processing as each read finishes traversing the pore. The authors may wish to clarify these differences.

---

## Round 0.2 · accepted · Accept

The revision significantly improves upon the the initial submission.

# ·

Basic reporting

All the comments were answered.

Experimental design

No particular comments

Validity of the findings

No particular comments

Additional comments

It would be nice if a new release of Krocus could be provided: the Github master has been significantly updated (e.g., the new output format added), but PyPI and Conda contain only the old release.